# Measurement report: Short-term variation of ammonia concentrations in an urban area increased by mist evaporation and emissions from a forest canopy with bird droppings

Kazuo Osada[1]

[1] GSES, Nagoya University, Furo-cho, Chikusa-ku, Nagoya 464-8601, Japan

*Correspondence to*: Kazuo Osada (kosada@nagoya-u.jp)

**Abstract.** Local meteorological conditions and natural and anthropogenic sources affect atmospheric $NH_3$ concentrations in urban areas. To investigate potential sources and processes of $NH_3$ variation in urban areas, hourly $NH_3$ and $NH_4^+$ concentrations were measured during November 2017 – October 2019 in Nagoya, a central

Japanese megacity. Average $NH_3$ concentrations are high in summer and low in winter. Daily minimum $NH_3$ concentrations are linearly correlated with daily minimum air temperatures. By contrast, daily maximum $NH_3$ concentrations increase exponentially with temperature, suggesting that different nighttime and daytime processes and air temperatures affect concentrations. Short-term increases of $NH_3$ concentrations of two types were examined closely. Infrequent but large increases (11 ppb for 2 hr) occurred after mist evaporation during daytime.

During two years of observations only one event of this magnitude was identified in Nagoya, although evaporation of mist and fog occurs frequently after rains. Also, short-term increases occur with a large morning peak in summer. Amplitudes of diurnal variation of $NH_3$ concentration (daily maximum minus minimum) were analyzed on days with non-wet and low-wind conditions: amplitudes were small (ca. 2 ppb) in winter, but they increased from early summer along with new leaf growth. Amplitudes peaked in summer (ca. 20 ppb) because of droppings

from hundreds of crows before roosting in trees on the campus. High daily maximum $NH_3$ concentrations were characterized by a rapid increase occurring 2–4 hr after local sunrise. In summer, peak $NH_3$ concentrations at around 8 a.m. in sunny weather were greater than in cloudy weather, suggesting that direct sunlight particularly boosts the morning peak. Daily and seasonal findings related to the morning peak imply that stomatal emission at the site causes the increase. Differences between daily amplitudes during the two summers was explained by

the different input amounts of reactive nitrogen from bird droppings and rain, suggesting that bird droppings, a temporary rich source of $NH_3$, affected the small forest canopy.

## 1 Introduction

Ammonia ($NH_3$) plays an important role in various atmospheric chemical processes (Behera et al., 2013). In fact, $NH_3$ is a major precursor of fine aerosol particles containing ammonium sulfate and ammonium nitrate (Seinfeld

and Pandis, 2016). In addition, aerosol particle acidity is modified by neutralization with $NH_3$ (e.g. Murphy et al., 2017; Song and Osada, 2020). Aerosol particles affect human health and climate. Therefore, reduction and control of its aerosol concentration are desirable under most circumstances (Dockery et al., 1993; IPCC, 2013). As an important gaseous precursor of aerosol particles, $NH_3$ sources and factors affecting concentrations have been studied for decades. Various natural and anthropogenic sources of $NH_3$ have been identified (Sutton et al., 2008; Behera et al., 2013). Although agricultural $NH_3$ sources such as domestic animals, and fertilizer loss are dominant emitters on a global scale, non-agricultural sources such as motor vehicles, industry, garbage, sewage, humans, and wild animals are also major sources, especially in urban areas (e.g., Perrino et al., 2002; Pandolfi et al., 2012; Reche et al., 2012; Sutton et al., 2000). For example, in the UK, Sutton et al. (2000) estimated non-agricultural $NH_3$ emissions as 19% of total emissions. As a non-agricultural source, seabirds were also widely recognized as important contributors (Sutton et al., 2000; Blackall et al., 2007; Riddick et al., 2012; Croft et al., 2016).

According to source–receptor analysis of atmospheric $NH_3$, the effective distance of a strong point source is mostly limited to a few kilometers (Asman et al., 1989; Hojito et al., 2006; Theobald et al., 2012; Shen et al., 2016). Agricultural facilities and seabird colonies acting as strong $NH_3$ sources are normally absent from densely populated urban areas. Therefore, a mixture of various small non-agricultural sources is expected to be the main contributor for local atmospheric $NH_3$, which potentially acts as a precursor of aerosol particles.

Three-way catalytic converters and selective catalytic reduction systems have been applied as after-treatment devices to reduce air pollutant emissions such as CO, hydrocarbons, and $NO_x$ in vehicular exhausts. However, exhaust from devices often contains $NH_3$ as a side product created under non-ideal conditions of after-treatments (Kean et al., 2009; Suarez-Bertoa et al., 2017). Vehicular emissions of $NH_3$ engender local and regional increases of ambient concentrations, especially during stagnant calm wind conditions in some megacities (Osada et al., 2019). In fact, garbage containers have been implicated as a more important contributor than sewage systems in Barcelona, Spain (Reche et al., 2012). Emissions from humans and pets have been implicated as a major nonagricultural urban source of $NH_3$ (Sutton et al., 2000). Recently, Hu et al. (2014) reported green spaces in downtown Toronto, Canada as a potential source of ambient $NH_3$ based on analyses of local, regional, and temporal variations of $NH_3$ concentrations. Similarly, Teng et al. (2017) pointed out the importance of $NH_3$ emissions from urban green spaces in Qingdao, a coastal urban area in northern China. Nevertheless, $NH_3$ emission processes from green spaces are not well known for urban environments. Green spaces in urban areas provide habitable environments for wild animals. Among wild animals found in urban areas, crows have adapted particularly well to city environments. As a consequence, crow populations have increased in many urban areas in recent years (Ueta et al., 2003; Vuorisalo et al., 2003). They form large roosts in scattered forests in urban areas and drop excreta from trees and wires to the ground during pre-roosting assembly and when resting in roosts. Nevertheless, no report has described a study of the potential of $NH_3$ emissions related to bird droppings in urban green areas.

Analysis of hourly concentrations in the atmosphere is useful to ascertain the sources and processes of ambient $NH_3$. For example, temporal correlation between vehicular exhaust species such as NOx, CO, and elemental carbon in urban area has been found for vehicular emissions of $NH_3$ (e.g. Perrino et al., 2002; Nowak et al., 2006; Osada et al., 2019). Moreover, temporal analyses have been made of $NH_3$ concentrations at grassland areas, which have revealed a link between morning peaks and dew formed on plant surfaces during the previous night (Wentworth et al., 2014; 2016). Hourly $NH_3$ measurements are a key technique to ascertain the bidirectional exchange of $NH_3$ through the canopy layer (e.g. Wyers and Erisman, 1998; Nemiz et al., 2004; Kruit et al., 2007; Hansen et al., 2013; Hrdina et al., 2019) because $NH_3$ transfer is governed by rapidly changing meteorological (sunlight availability, temperature, relative humidity, etc.) and plant physiological (stoma opening and closing, etc.) parameters (Schjoerring et al., 1998; 2000). In fact, $NH_3$ exchange between plants and ambient air occur mainly through stoma when they open during daytime for photosynthesis (Farquhar et al., 1980). Therefore, the degree and direction of $NH_3$ exchange are expected to vary diurnally, highlighting the importance of hourly measurements of related parameters.

To investigate potential sources and processes controlling the variation of $NH_3$ concentration, hourly data of $NH_3$ and $NH_4^+$ were recorded from November 2017 through October 2019 in Nagoya, central Japan. The data were analyzed by particularly examining various time scales and the amplitude of diurnal variation in relation to potential reactive nitrogen sources and plant physiology near the site. These data were expected to elucidate effects of large amounts of bird droppings on ambient $NH_3$ concentrations in urban areas with scattered forests.

## 2 Observation

Atmospheric observations were made at Nagoya University in Nagoya city located in the central area of Honshu Island of Japan (Fig. 1a). This industrial area with a busy port is located about 10 km southwest from the campus of Nagoya University (Fig. 1b). The Nagoya city population is about 2.3 million. Despite the large city population engaged in various industrial activities, air pollution levels are not so high. Recent annual mean $PM_{2.5}$ concentrations have been approximately 12 $\mu g/m^3$ (Nagoya City, 2019: http://www.city.nagoya.jp/kankyo/page/0000117927.html). The nearest agricultural activities (farming land) are done about 4 km southeast from the campus. Garbage collection in the city requires 1) that burnable waste including food waste and other materials be packed into predesignated plastic bags and 2) that the garbage bags be put out in a specified collection place by 8:00 a.m. on a regular (twice per week) collection day, preventing unnecessary $NH_3$ emissions during garbage collection. However, the garbage bags might be pecked by crows when deterrents to bird pecking are insufficient, presenting a possible food supply for the adaptation of omnivorous animals, such as crows, in urban areas (Kurosawa et al., 2003). The observation site is located within

the campus. Therefore, the effects of residential garbage are expected to be small. The annual mean air temperature in Nagoya is 15.8°C; its annual mean rain amount is about 1540 mm (Japan Meteorological Agency: http://www.jma.go.jp/jma/index.html). Seasons in Nagoya have warm-humid summers with southern winds from the Pacific Ocean and cold-dry winters with winds from the northwest, originating from continental Eurasia.

Measurements of $NH_x$ ($NH_3$ and $NH_4^+$ in fine particles) were taken at Nagoya University (35.16°N, 136.97°E), located in an eastern residential area of Nagoya city. Meteorological data (air temperature, relative humidity, etc.) were obtained from the Nagoya Local Meteorological Observatory, ca. 2 km north from Nagoya University (data available from https://www.jma.go.jp/jma/index.html). For this study, NOx and CO concentrations were observed at the Nagoya national air pollution monitoring site located ca. 2 km north from Nagoya University (data available

from http://soramame.taiki.go.jp/).

      The equipment used for $NH_x$ measurements was set up in a room on the seventh floor (ca. 26 m above the ground) of the Environmental Studies Hall on the main campus of Nagoya University. The northeastern side of the building faced upslope with a small forest with mixed deciduous and evergreen trees (Fig. 1c). As Fig. 1c shows, the seventh floor height is almost equal to the height of the forest canopy growing about 40 m away on

the northeastern slope. Scattered trees and buildings are located on the other side of the hall. Hourly measurements of $NH_x$ were conducted using a semi-continuous microflow analytical system (MF-NH3A; Kimoto Electric Co. Ltd.) described in an earlier report (Osada et al., 2011). Two identical sampling lines were used to differentiate total ammonium ($NH_3$ and $NH_4^+$) and particulate $NH_4^+$ alone after removal by a $H_3PO_4$-coated denuder. The sample air flow rate of the $NH_x$ system was 1 L min$^{-1}$. After passing an impactor (about 2 μm cut-off diameter)

and an inner frosted glass tube (one coated, the other uncoated, both of 3 mm inner diameter and 50 cm long), pure water droplets were added immediately to the sample air at 100 μl min$^{-1}$. The collection efficiency of the system was greater than 95% for the conditions used in this study. The equilibrated sample water was analyzed respectively using a microflow fluorescence analyzer to quantify $NH_4^+$ in the lines of $NH_4^+$ and total ammonium. The $NH_3$ concentration was calculated based on their difference. The temporal resolution was ca. 30 min for one

pair of $NH_4^+$ and total ammonium measurements. The detection limit of $NH_3$ concentration was about 0.1 ppbv (Osada et al., 2011) under stable atmospheric $NH_3$ and $NH_4^+$ concentrations. Equivalence of two sample lines and the span of the calibration slope was checked monthly using $NH_3$ standard gas at about 4 ppb diluted from 100 ppm (Taiyo Nippon Sanso Corp.). The $NH_x$ system was calibrated monthly using a standard $NH_4^+$ solution prepared from a certified 1000 ppm solution (Fujifilm Wako Pure Chemical Corp.).

**3 Results and Discussion**

**3.1 Diurnal variation during summer and winter**

Results of measurements taken in summer (July, 2018; Fig. 2) and winter (December, 2018; Fig. 3) are presented to explain the relation between $NH_3$ concentrations and other parameters. In summer (Fig. 2), the Pacific high-pressure system dominates the Japan archipelago, engendering continuous good weather with land−sea breeze cycles: south−southwest winds during afternoon and north−northeast winds after midnight to early morning. When good weather continued, for example of 14–24 July, regular diurnal variations were found for air temperature and wind speed: high in the afternoon and low in the midnight to early morning. During this period, the difference between maximum and minimum $NH_3$ concentrations was large: from nearly 10 to more than 20 ppb. However, the $NH_3$ concentration dropped to a few parts per billion. It remained constantly low for the duration of the rain with small diurnal variation because, during 4–7 July, the Baiu front (East Asian rainy front) was active near the site. A similar tendency of low $NH_3$ concentration during rainy days was found during observations, as reported from other studies (Roelle and Aneja, 2002; Ellis et al., 2011). Increased contents of soil pore water dilute $NH_4^+$ in liquid phase and inhibit evaporation as $NH_3$. Furthermore, wet surfaces of cuticular leaves absorb ambient $NH_3$ under high relative humidity during rain. Moreover, $NH_3$ concentrations were low during days of higher wind, such as 23 July. The NOx concentrations in July were mostly below 0.02 ppm with no clear correlation with $NH_3$ variation.

Figure 3 presents data of $NH_3$ concentrations and other parameters for December 2018. The maximum value of the vertical axis of $NH_3$ concentrations is 15 ppb, which is half of that in Fig. 2. In winter, the amplitude of the diurnal variation of $NH_3$ concentration was much smaller than in summer. Diurnal wind speed was variable in winter because of the lack of sea breeze circulation. In contrast to summer, wind speed dependence of $NH_3$ concentrations is clearer, showing higher concentrations under calm winds and lower concentrations under strong wind conditions, because local sources more strongly affect local conditions under stagnant air conditions. With low winds during winter, a surface inversion layer often developed in the lower atmosphere, preventing vertical diffusion of locally emitted pollutants (e.g., Kukkonen et al., 2005, Osada et al., 2019). It is particularly interesting that temporal variations of daily minimum $NH_3$ concentrations in winter roughly follow the day to day variation of the daily minimum temperature. For example, a higher minimum $NH_3$ concentration of about 3 ppb during 2−4 December decreased to ca. 1 ppb around 10 December, which corresponds to a decreasing trend of high to low air temperatures for these days.

In contrast to modest variation found for July, NOx concentrations in December showed large variation: concentrations were frequently found to be greater than 0.05 ppm during calm winds. In Fig. 3, the concentration peaks on 3–4, 11, 19–23, and 25–26 December were associated with low winds. High NOx concentrations were strongly associated with high CO concentrations (Suppl. Figure 1), suggesting strong contributions of vehicular

emissions for both species. Similarly to NOx, the NH$_3$ concentrations increased under low winds. The similarity of NH$_3$ temporal variation with NOx suggests that motor vehicle emissions partly contribute to ambient NH$_3$ concentration in winter (Osada et al., 2019). However, neither NH$_3$ nor NH$_x$ (NH$_3$ and NH$_4^+$) showed strong correlation with NOx (Suppl. Figure 1), suggesting that a source other than vehicular emissions contributes more to NH$_3$ concentrations in winter.

Figure 4 depicts average diurnal variations of NH$_3$ and NH$_4^+$ concentrations, air temperature, solar radiation, relative humidity (RH), wind speed (WS), NOx and CO concentrations from December 2017 through September 2019 for every 3 months. For NH$_3$ variation, seasonal change was readily apparent. Broad and modest maximum of NH$_3$ concentrations were observed at around noon in December and March. By contrast, in June and September, sharp peaks at around 8 o'clock in the morning were superimposed on the broad and modest variation during colder season. The start timing of the NH$_3$ increase in the morning during the warm season was similar to the hours of rising air temperature and dropping relative humidity. In contrast to diurnal variations of NOx and CO concentrations, NH$_3$ concentrations in December and March showed no large morning peak during rush hour between 7 and 9 am, suggesting that contributions of NH$_3$ from motor vehicular emissions are apparently limited to those days under calm wind conditions, even in December.

Diurnal variation of NH$_4^+$ concentration was nearly flat for all examples portrayed in Fig. 4. Regarding the relation between NH$_4^+$ concentrations and air temperature, dissociation of particulate NH$_4$NO$_3$ in the atmosphere is known to be strongly related to temperature: partitioning toward gas phase is favored under higher air temperatures (Mozurkewich, 1993). However, the NH$_4^+$ concentrations in Figure 4 were not simply due to diurnal temperature variations. Therefore, NH$_3$ increases at around noon during the cold season are not attributable to dissociation of NH$_4$NO$_3$. Solar radiation showed higher values around noon. Wind speed was low from midnight to the morning and high during afternoon and early evening.

Diurnal variation of NH$_3$ concentration is controlled mainly by time variations of 1) atmospheric boundary layer dynamics, 2) dry deposition to cuticular of vegetation and other surfaces, and 3) local emission strength (Sutton et al., 1995; Saylor et al., 2010; Hrdina et al., 2019). First, the boundary layer height increases with surface heating by solar radiation, leading to greater dilution of the NH$_3$ concentration at noon. If this is the case, then diurnal variation of NH$_3$ resembles those of NOx and CO. However, it appears to be different in Fig. 4, suggesting dominance of other factors. Second, dry deposition velocity of NH$_3$ varies with wetness of the surfaces. Diurnal variation of RH shows minimum values around noon, implying that the NH$_3$ dry deposition in daytime is greater than that during nighttime. Evaporation of dew and that from wet surfaces is also a potential source of NH$_3$, as discussed in greater detail in section 3.2. Lastly, time variation of local emissions is important, especially for a low wind conditions. Rush-hour emissions from motor vehicles were not the major contributor, as discussed earlier. To seek an effective source creating the morning peak, diurnal variations were analyzed again for days selected by weather condition (more specifically duration of sunshine) as presented in Fig. 2.

Figure 5 depicts the effects of sunshine on the morning peak of $NH_3$ concentrations during July and August for 2018 and 2019. The left column represents fine weather of days receiving direct sunlight longer than 10 hr (clear sky for most of the day). The right column shows cloudy weather with sunshine duration shorter than 5 hr and daily rain of less than 3 mm. The $NH_3$ concentration showed a strong peak at 8 a.m. under fine weather, although the peak concentration was slightly lower in 2019. By contrast, a strong peak was absent on the mornings of cloudy days. This trend with direct solar radiation suggests that the morning peak is attributed to plant physiological activities such as photosynthesis or stoma opening. The importance of stomatal $NH_3$ emissions is discussed further in section 3.3.

## 3.2 Peak after mist evaporation

Figure 6 presents an example of a sudden increase from 2 ppb to 13 ppb of $NH_3$ concentration during 2 hr associated with drying mist after rain on 15 November, 2017. This magnitude of $NH_3$ increase after rain was only rarely observed during the two year study period. Mist is defined as reduced horizontal visibility to 1–10 km because of suspended water droplets in the atmosphere. In Nagoya, mist often occurs before or after rain. In this case, rain ceased in the early evening of November 14, but the mist persisted until 10 am on 15 November. Associated with relative humidity dropping sharply from ca. 90% at 10 am to ca. 40% at noon, the mist disappeared. The $NH_3$ concentration increased abruptly due to temporal variation of relative humidity (RH). Slight enhancement of $NH_3$ concentration after the rain has been described in some reports of the literature (Roelle and Aneja, 2002; Ellis et al., 2011). Those reports presented discussion of the hypothesis on enhancement associated with the combination of an increase in the ammoniacal nitrogen concentration in the soil and diffusion from the soil to air after the drying pore solution. However, this process requires more time after rain cessation to decrease soil moisture; then, it is too slow to raise the atmospheric $NH_3$ concentration. In contrast, Wentworth et al. (2014, 2016) reported that a rapid increase of $NH_3$ in the morning was attributed to evaporation of dew containing high concentrations of $NH_4^+$ with nearly neutral pH. They reported that the amount of $NH_3$ volatilized from dew is governed by the ionic composition (excess amount of $NH_4^+$ rather than forming less-volatile salts from constituents) and pH (gas–aqueous partitioning and chemical equilibria in solution). Although the magnitude of the morning increase reported by Wentworth et al. (2016) was less than half that of the present case, the timing and quickness of $NH_3$ increase were similarly to those depicted in Fig. 6, suggesting that mist droplet evaporation is a major source process for high $NH_3$. A similar rapid $NH_3$ increase up to 15 ppb during 4 hr was observed earlier on 11 December, 2015 in Nagoya (Osada et al., 2018). According to an acid rain report in 2017 published by the Nagoya City Institute of Environmental Sciences (NCIES, 2018), the volume weighted mean pH of the weekly collected rain sample from 13–20 November, 2017 was 6.0. Based on major ionic data reported for the sample, Frac ($NH_4^+$) proposed by Wentworth et al. (2016) was estimated as 0.14, which suggests the possibility

of NH$_3$ evaporation after drying of the rain. However, rain was observed twice during the sampling period: on 14 November (9 mm) and 18 November (16.5 mm). Unfortunately, the chemical composition of rain on 14 November was not known, so further discussion is difficult. Instead, the duration of mist after the rain might be an important factor to form a favorable composition for releasing NH$_3$. In fact, the duration of mist after the rain

was unusually long (16 hr). Therefore, ambient NH$_3$ during the mist was able to dissolve into the droplets better. Subsequently, a large amount of NH$_3$ was released from mist droplets after evaporation, engendering a spike of the NH$_3$ concentration.

### 3.3 Emissions from the tree canopy surrounding the site: relation to bird droppings

Figure 7 presents results of data analysis of NH$_3$ and NH$_4^+$ concentrations. Monthly NH$_x$ (NH$_3$ + NH$_4^+$) concentrations (Fig. 7a) are high in summer and low in winter. Daily and monthly NH$_4^+$ (Fig. 7b) concentrations show higher values from spring to summer and lower in fall and winter, whereas daily and monthly NH$_3$ concentrations (Fig. 7c) depict clear seasonal variation: high in summer (maximum in August) and low in winter (minimum in January). Amplitudes of seasonal variations of NH$_3$ are larger than those of NH$_4^+$. Consequently,

NH$_3$ controls the seasonal variation of the NH$_3$ fraction (Fig. 7b): ca. 60–75% in late summer (maximum in August–September) and 40–50% in late winter to early spring (minimum in February and March). For the NH$_3$ concentration, the monthly minimum (1.6 ppb) in January 2018 was almost equal to that (1.7 ppb) in January 2019, although the monthly maximum (7.0 ppb) in August 2018 was higher than that (4.9 ppb) in August 2019. Furthermore, day-to-day variation was greater in 2018 than in 2019. To examine the relation with source factors,

hourly NH$_3$ concentrations were analyzed for two subjects: the daily minimum and the diurnal range (maximum minus minimum) under dry (RH <70%) and weak wind (<3 m s$^{-1}$) conditions for days with both daily mean values. Reasons for the meteorological limitations were the following. Wet surfaces on building walls, litter, soil, and leaves can act as a NH$_4^+$ reservoir, which might change ambient NH$_3$ concentration shortly after evaporation. To avoid this effect, the daily average of relative humidity was set to below 70% for extraction as "non-wet days".

As Figs. 2 and 3 show, the wind speed exhibited a strong effect on local source dilution. Therefore, a day of weak wind was selected to illustrate a stronger effect of the local source.

The daily minimum NH$_3$ concentrations are shown together with the daily minimum air temperature (Fig. 7d). As briefly described earlier for Fig. 3, day-to-day variation of daily minimum NH$_3$ concentration in December covaries with the baseline trend in daily minimum temperature. Analogously to this, the seasonal variation of

255 daily minimum NH$_3$ concentrations follows closely the seasonal variation of the daily minimum temperature: high in summer with larger variation, and low in winter with less variation during the month. Monthly averages of daily minimum NH$_3$ concentrations were higher in summer of 2018 (ca. 4 ppb) than those of 2019 (ca. 2.8 ppb), but almost identical values (0.7−1 ppb) were obtained for the respective winters. Daily minimum values of

concentration and temperature were usually observed in the early morning before sunrise (Fig. 4). Although vertical profiles of $NH_3$ concentration and meteorological parameters were not available, the ambient $NH_3$ at the time of daily minimum is presumably derived from local origin under low wind. The $NH_3$ in the air is equilibrated with the local surfaces of plants and soils. Stomata of plants do not open before sunrise. Therefore, stomatal gas exchange is expected to be negligible before sunrise. Furthermore, plant surfaces are less effective as $NH_4^+$ reservoirs because RH <70%. However, pore water or moisture in soil can remain. That moisture might act as a bidirectional exchange source of $NH_3$. For $NH_3$ equilibrium between soil pore water and air, known as a compensation point, important parameters aside from the atmospheric $NH_3$ level include the temperature, pH, and $NH_4^+$ concentrations in the solution (Farquhar et al., 1980). We discuss more details related to this point later using Fig. 8 b.

Another point is analysis of the amplitude of diurnal variation as the difference between the maximum and minimum of the day, denoted as daily max−min (Fig. 7e). As anticipated from the difference between summer and winter in daily $NH_3$ variations portrayed in Fig. 4, the daily max−min values were larger in warm months than in cold months. The daily max−min values were found to be quite large in summer (12.8 ppb as averages during July–August in 2018 and 9.2 ppb in 2019) and small in winter (2.3 ppb as the average during January–February in 2018 and 2.2 ppb in 2019). The increase of the daily max−min values began around May and ended gradually in September for these years. The timing of the start and the end for the large amplitude implies a connection with the leaf growth stage of deciduous trees around the site. Although evergreen trees (*Quercus glauca*, *Pinus*, *Machilus thunbergii*, etc.) are mixed, deciduous trees (*Quercus variabilis*, *Zelkova serrata*, Japanese cherry, *Liquidambar*, *Metasequoia glyptostroboides*, *Aphananthe aspera*, etc.) grow in the small forest (ca. 380 × 30 m) near the site. Figures 1c and 1d respectively show the tree belt and autumn leaves at the front of the building.

To study the emission potential of $NH_3$ from various reservoirs (i.e. apoplastic fluid of plants and soil pore water), the compensation point model is applied for comparison by estimation from temperature and thermodynamic equilibrium among various surfaces and the atmosphere. The compensation point ($X$) is predicted as the following.

$$X = \frac{161500}{T} \exp\left(-\frac{10380}{T}\right)\Gamma \qquad (1)$$

Therein, $T$ denotes the surface reservoir temperature in Kelvin, $\Gamma$ represents the emission potential equal to the concentration ratio between $[NH_4^+]$ and $[H^+]$ in the surface reservoir ($\Gamma = [NH_4^+]/[H^+]$), and $X$ is given in units of parts per billion or in nanamoles per mole (Nemitz et al., 2004). Greater emission potential represents higher

equilibrium $NH_3$ concentration with the surface for the same temperature. Higher temperature raises the equilibrium $NH_3$ concentration present as gas phase.

Apoplastic fluid in stomata of plants and pore water in soil are assumed as the major reservoirs of $NH_4^+$. To equilibrate apoplastic fluid with ambient atmosphere, stomata must be opened, which is regulated by the plant physiology related to photosynthesis. Consequently, daily maximum $NH_3$ concentrations were observed at about 2−4 hr after sunrise. In other words, the initial stage of stomata opening is synchronized well with the timing of the morning increase. The daily maximum $NH_3$ concentrations are shown versus the average air temperature of the day (Fig. 8a). Leaf temperatures were not measured for this study. The ambient temperature was used as a surrogate of leaf temperature. In Fig. 8a, two hypothetical compensation curves are also shown using $\Gamma$ of 1500 and 200. At air temperatures higher than about 10–15°C, most observed data are shown between these two curves, suggesting that the $\Gamma$ of the forest canopy around the site is in the range of 200–1500. According to the literature related to compensation points (Massad et al., 2010; Zhang et al., 2010; Hrdina et al., 2019), $\Gamma$ was several tenths to $10^5$ depending on the type of ground, vegetation, and richness of reactive nitrogen available for plants. For stomatal emission potential of $NH_3$, the range of 300–3000 for trees of deciduous and evergreen forests was proposed by Zhang et al. (2010). A similar range of values was listed by Massad et al. (2010).

Furthermore, daily minimum $NH_3$ concentrations are shown versus the minimum air temperature of the day (Fig. 8b). As described earlier, the condition observed for daily minimum $NH_3$ concentration is related to the emission potential from soil around the site because stomatal emissions are negligible. Although soil temperatures were not measured for this study, the minimum air temperature was used as a surrogate for nighttime soil temperatures. In Fig. 8b, two hypothetical compensation curves are shown using $\Gamma$ of 500 and 200. Observed data were in the range of the hypothetical compensation curves only for minimum air temperatures higher than 20°C. Below 20°C, most observed data were over the curve for $\Gamma$ of 500. Two possibilities can be considered for these relations. One is that higher $\Gamma$ for soil is responsible for winter because litter from deciduous trees can be decomposed by microbial activity. In addition, subsequent $NH_4^+$ production raised $\Gamma$ higher than 500. Another possibility is that the contribution from vehicular emissions was enhanced by stagnant air pollution in winter (Yamagami et al., 2019).

As shown separately in Fig. 8, concentrations of $NH_3$ in summer of 2018 were higher than those of 2019 for comparison with the same temperature. $\Gamma$ for the canopy of a site varies with various parameters such as seasonal variation of plants' stages of growth and supply of reactive nitrogen (Schjoerring et al., 1998; 2000; Massad et al., 2010). Senescent and mature leaves have high potential for $NH_3$ emissions (Mattsson and Scjoerring, 2003). For deciduous trees related to the present study, new leaves start to grow in April; they mature after June. They turn red in November. The duration of active leaves of deciduous trees roughly accords with the season of the higher daily max–min portrayed in Fig. 7e. However, the values of the daily max–min in summer differed between

2018 and 2019, with no great change of trees on the campus. As discussed in section 3.1, dry deposition of $NH_3$ is controlled by surface conditions of the soil and cuticular plant surfaces. In the analyses portrayed in Figs. 7 and 8, drier days were selected, excluding the complexity of wet processes for $NH_3$ exchange between air and various surfaces. In addition, leaf conditions of trees and weeds were almost identical for these years, suggesting that conditions of $NH_3$ dry deposition did not change. In addition, micrometeorological factors govern the transfer velocity that ultimately determines the magnitude of the $NH_3$ exchange. Unfortunately, meteorological data required for estimating the transfer velocity were not available in this study. Further data for flux estimations are necessary to evaluate $NH_3$ exchange in urban areas. Nonetheless, important suggestions can be made for potential source variation at the site. As input to the system, the amount of reactive nitrogen brought by wet deposition (rain) varies slightly year-to-year. According to annual reports of acid rain (NCIES, 2019; 2020), monthly average wet depositions of $NO_3^-$ during May–September were 2.3 mmol $m^{-2}$ in 2018 and 2.0 mmol $m^{-2}$ in 2019. Similarly, monthly average wet depositions of $NH_4^+$ during May–September were 2.9 mmol $m^{-2}$ in 2018 and 2.4 mmol $m^{-2}$ in 2019. Wet deposition of these species during warm months was slightly (ca. 15%) higher in 2018 than in 2019. However, the observed differences (ca. 30%) in the average daily max–min between 2018 and 2019 were almost double those in wet deposition, requiring more input to explain yearly discrepancies.

To seek more input to the system, the importance of bird droppings at the site is discussed below. From June or July through September or October, rooftops of buildings and trees on the campus are used frequently by more than several hundred crows for pre-roosting assembly or flight line assembly in early evening before going to roost, located on or presumably near the campus. Normally, a murder of crows stays a short time (mostly less than 2 hr). They then leave to their primary roost area (Nakamura, 2004). More crows gathered in the murder in summer in 2018 than in 2019, which is regarded as the number density of white fecal remains in photographs under trees at the front of the building (see Appendix Photograph 1).

Bird droppings are rich in reactive nitrogen: nitrogen contents in dry weight of droppings are 3.5% for chicken (Nakamura and Yuyama, 2005) and 4.7% for crows (Fujita and Koike, 2007). The major reactive nitrogen of bird droppings is uric acid, which is readily transformed into $NH_4^+$ by microbial activity in the soil. It is later incorporated and used by plants through roots. To evaluate the effects of bird droppings at the site, the flux of reactive nitrogen added by bird droppings over the unit area ($Flux_{bd}$, mol $m^{-2}$ $day^{-1}$) is estimated as shown below.

$$Flux_{bd} = \frac{Freq \cdot W \cdot R}{14} \qquad (2)$$

In that equation, $Freq.$ (number $m^{-2}$ $day^{-1}$) represents the input frequency of excreta shot per day over unit area, $W$ (g $shot^{-1}$) stands for the dry weight of excreta per excreta shot, $R$ (%) denotes the nitrogen content per dry excreta weight, and 14 is the atomic weight of nitrogen for conversion. For simplicity, the following values are

used to estimate $Flux_{bd}$: $Freq$ is once per day per square meter, $W$ is 1 g per shot, and $R$ is 4.7% (Fujita and Koike, 2007). Evaluating the relevance to the assumptions is difficult, but it is believed to be the best guess from the dropping situations observed around the building (Appendix Photograph 1). The estimated result is 3.4 mmol m$^{-2}$ day$^{-1}$, which is converted to ca. 100 mmol m$^{-2}$ month$^{-1}$. This value is nearly 40 times higher than the NH$_4^+$ flux by rain. $Flux_{bd}$ includes large uncertainty depending on the number of crows gathered and their behavior on the campus. However, it is useful for comparison with reactive nitrogen flux by rain. Even assuming $Freq$ as one-tenth of the initial assumption above (0.1 number m$^{-2}$ day$^{-1}$), $Flux_{bd}$ is still larger than the NH$_4^+$ flux by rain. In this study, the dense area of bird droppings was not so large on the campus. However, the excess inputs of reactive nitrogen brought by crows to a small area might engender strong local emissions of NH$_3$ from the soil and through the forest canopy. Indeed, Fujita and Koike (2007) pointed out that jungle crows brought substantial amounts of nutrients to their roost of fragmented forests in an urban area. Populations of crows and the distribution of crow roosts vary with food availability and trees for sleeping and breeding. Crows have adapted well to urban areas. For that reason, their populations are often increasing in urban areas worldwide (Ueta et al., 2003; Vuorisalo et al., 2003). Through the increase of bird droppings, reactive nitrogen in urban small forests is oversaturated for tree growth and emitted excess nitrogen as NH$_3$ from the tree canopy. Vegetation in urban environments tends to catch and concentrate gaseous and particulate reactive nitrogen pollutants and to supply them to the ground surface (Decina et al., 2020). The present study yielded a particularly important point: this process is bi-directional for NH$_3$.

**4 Summary and Conclusions**

Hourly measurements of NH$_3$ and NH$_4^+$ were conducted from November 2017 through October 2019 in Nagoya, central Japan. Monthly average NH$_3$ concentrations were high in summer (7.0 ppb and 4.9 ppb, respectively, for August in 2018 and 2019) and low in winter (1.6 ppb and 1.7 ppb for January 2018 and 2019, respectively). During the study period, a surge event (11 ppb during 2 hr) was observed after mist evaporation during daytime, which was very rare at Nagoya, even though evaporation of mist or fog droplets is expected to be frequent after rain. A plausible condition of the surge event was discussed in terms of the composition and pH of rain. The amplitude of diurnal variation of NH$_3$ concentration (daily maximum minus minimum) was small (ca. 2 ppb) in winter and large (ca 10 ppb) in summer. The daily max–min increased from late spring synchronized with new leaf growth and peaked in summer during intense addition of droppings from hundreds of crows assembled on trees and rooftops near the site before going to their roosts. Large diurnal variation of NH$_3$ concentration was characterized by a peak at 2–4 hr after sunrise. In summer, the peak NH$_3$ concentration at around 8 a.m. under fine weather was larger than that under cloudy weather, suggesting that receiving direct sunlight is important for

boosting the morning peak. The timing of seasonal and daily increases of the morning $NH_3$ peak imply that
reactive nitrogen inputs from crow droppings and rain increased $NH_3$ emissions from the tree canopy. Preliminary
estimates suggest that reactive nitrogen input by crow droppings was greater than the effect of wet deposition.
Therefore, crow populations are increasing in some urban areas through adaptation. Reactive nitrogen supplied
by crow droppings might become an increasingly important source of $NH_3$ emissions in urban areas.

*Data availability.* The $NH_x$ and other data used in Figs 2–8 are available at NAGOYA Repository:
http://hdl.handle.net/2237/00032615.

*Supplement.* The supplement related to this article is available online at:

*Author contributions.* KO conducted all of this research.

*Competing interests.* The author has no conflict of interest related to this report or the study it describes.

## 5 Acknowledgments

This research was supported by the Environment Research and Technology Development Fund (5-1604) of the
Environmental Restoration and Conservation Agency. The author thanks Dr. Y. Hirano for discussion of soil pH
and reactive nitrogen from bird colonies. Constructive comments from two anonymous referees are also greatly
appreciated.

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

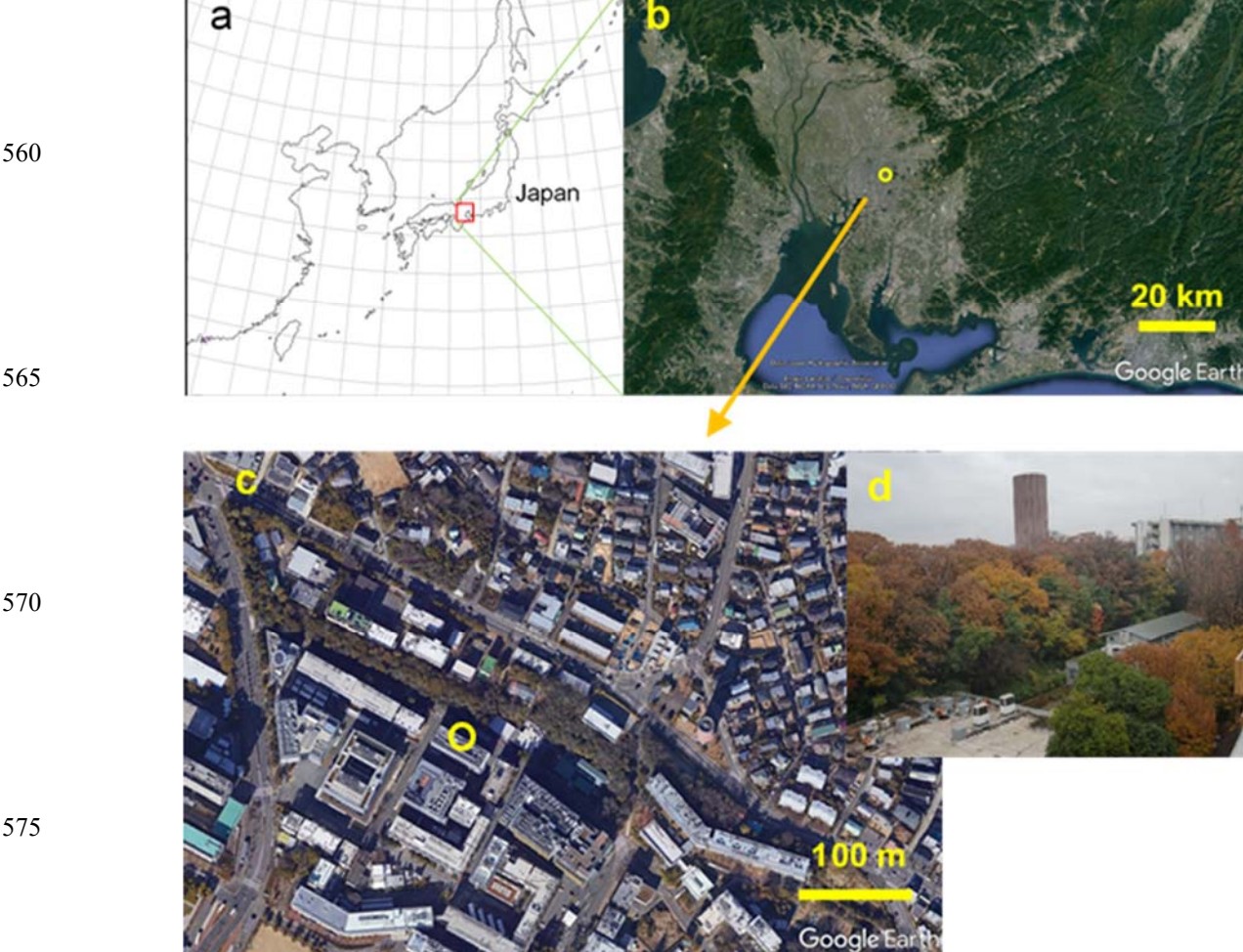





Figure 1(a) Map of the area around the observation site. (b) Satellite image (from © Google Earth) near Nagoya University (open circles: 35.67°N, 139.83°E) in Nagoya, Japan. A local meteorological station (air temperature, relative humidity, rain, wind speed and direction, solar radiation, weather records) and a national air pollution monitoring site (NOx and CO) are located at about 2 km north of the site. (c) © Google Earth close-up image of the campus. $NH_x$ measurements were taken at the Environmental Studies Hall (open circle in c). (d) Outside view from the seventh floor (26 m above the ground) to the northeast taken in early December.



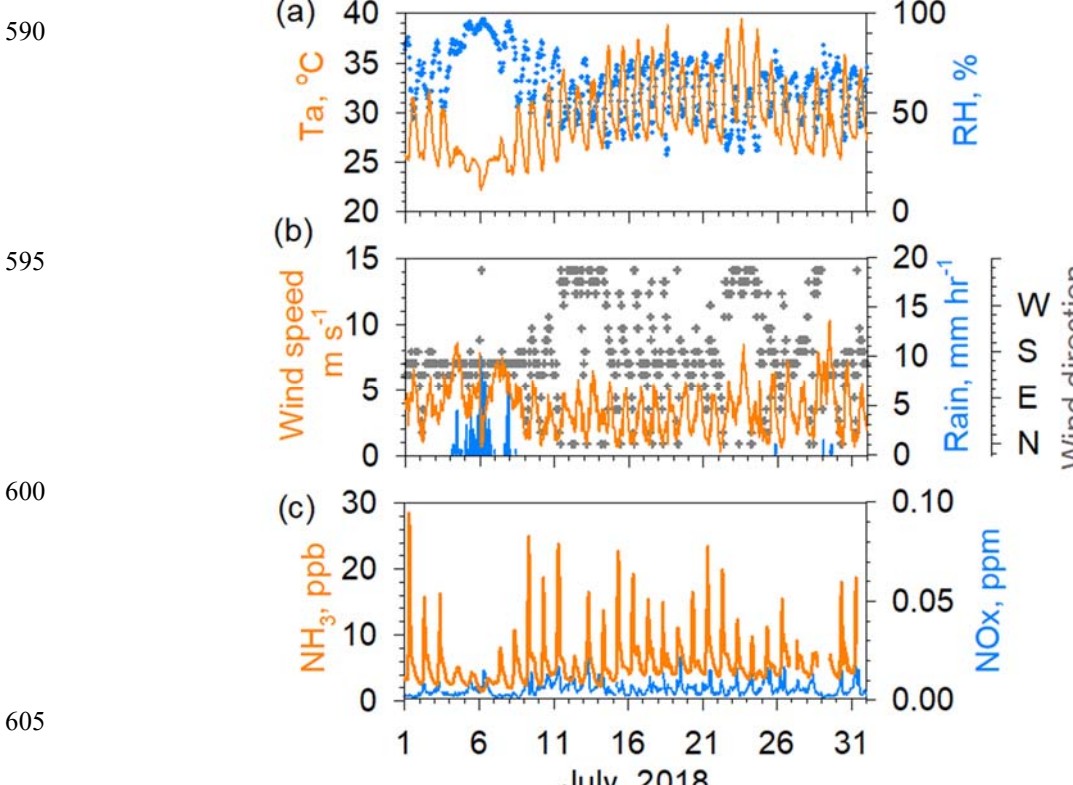

 Figure 2 a) Temperature (left axis) and relative humidity (right axis); b) wind speed, rain rate and wind direction; c) $NH_3$ and $NO_2$ concentrations in Nagoya during July 2018.

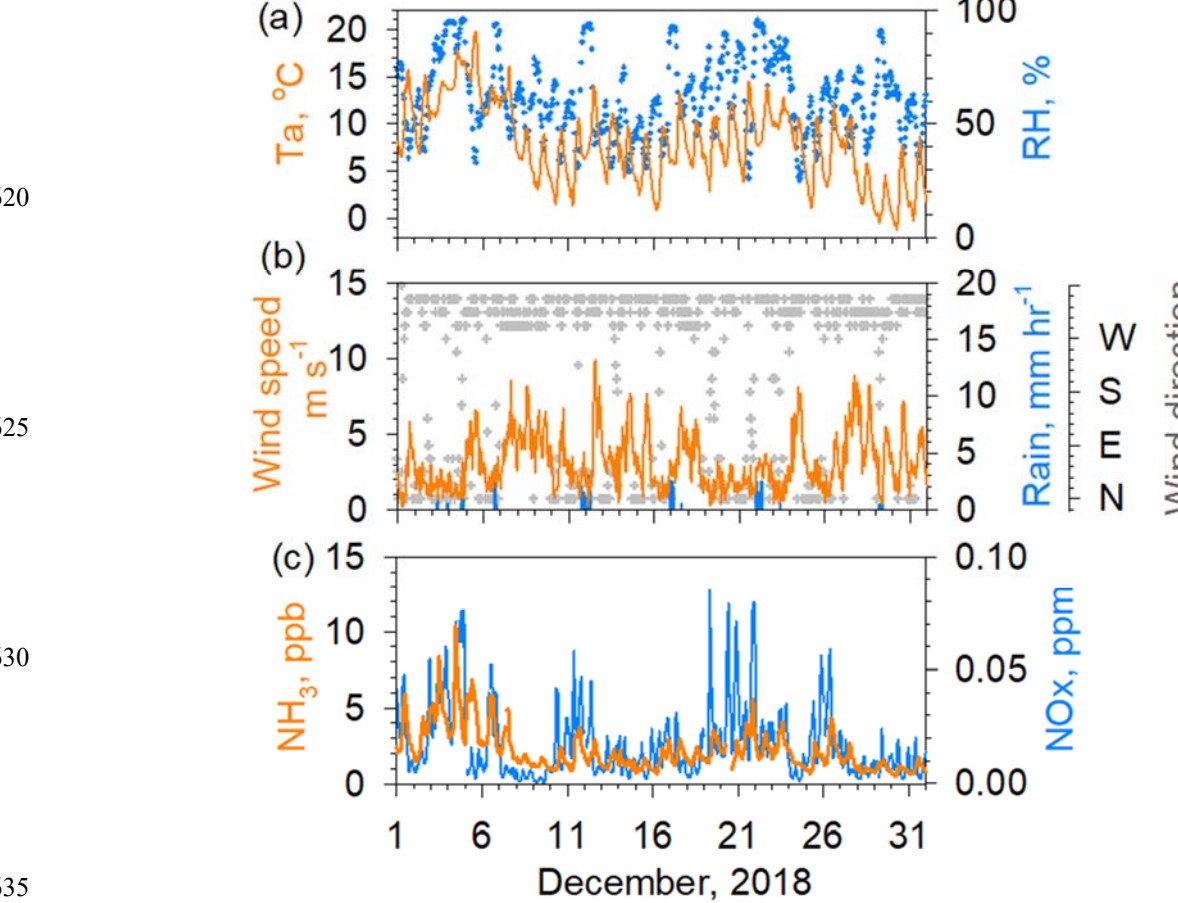

Figure 3 Same as Fig. 2, but for results obtained during December 2018. The maximum of the vertical axis for NH$_3$ is reduced to 15 ppb.

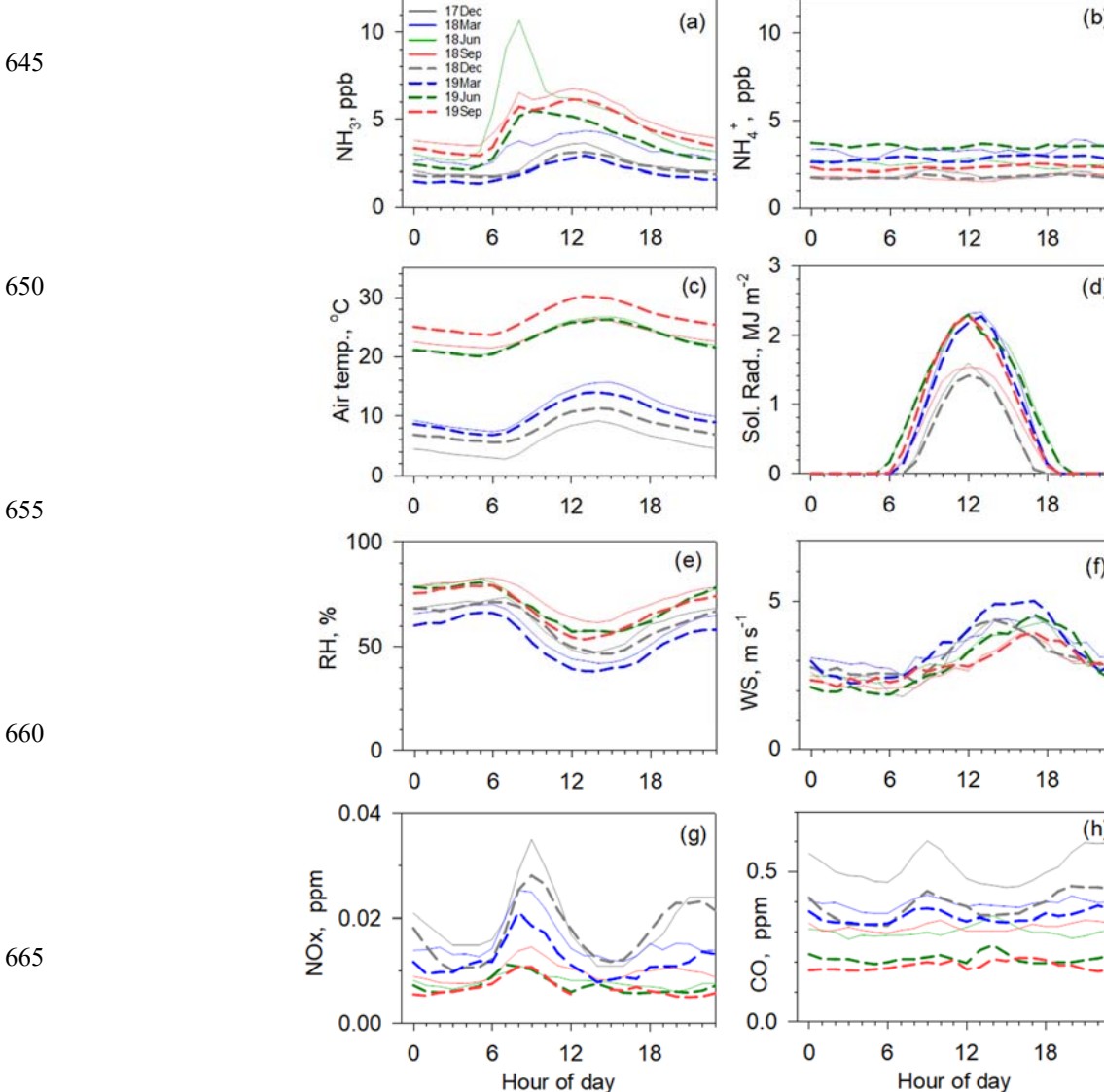

Figure 4 Hourly averages of various concentrations and meteorological parameters for several months: December (fine gray), 2017; March (fine blue), June (fine green), September (fine orange), and December (dotted gray), in 2018; March (dotted blue), June (dotted green), September (dotted orange), 2019; a) $NH_3$, b) $NH_4^+$, c) air temperature, d) solar radiation, e) relative humidity, f) wind speed, g) NOx, h) CO.

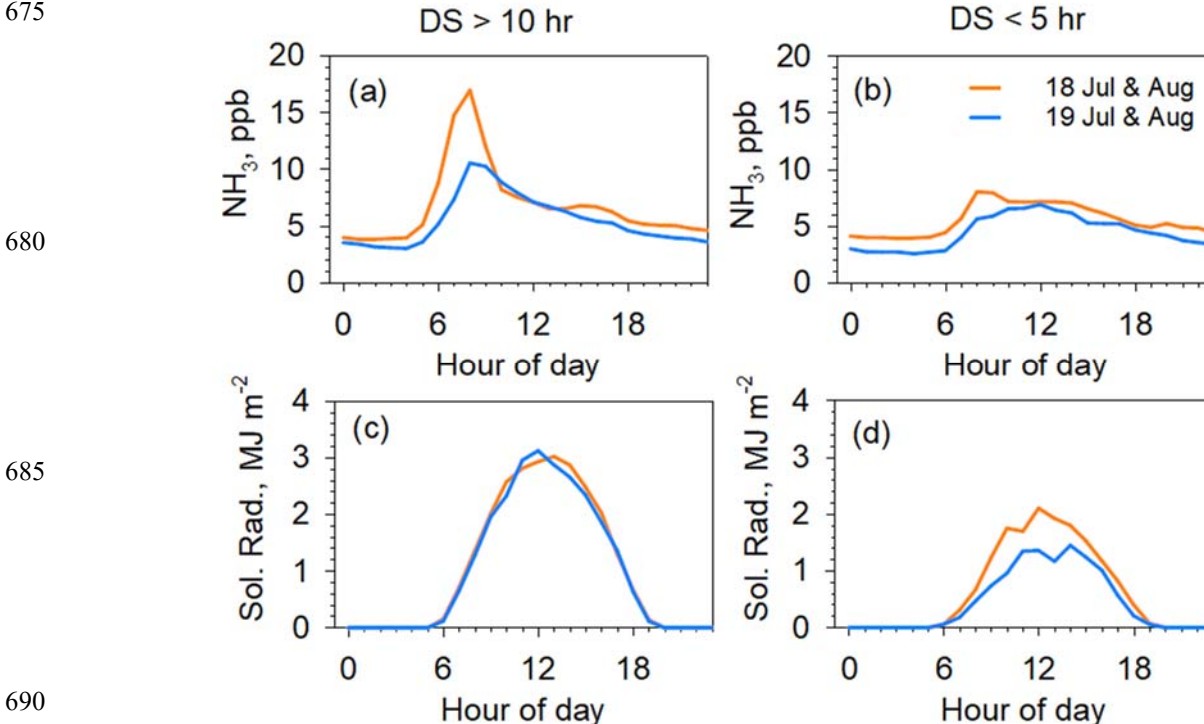

Figure 5 Hourly average of $NH_3$ concentration (a and b) and solar radiation (c and d) during July–August in 2018 (orange) and 2019 (blue). Left column (a and c): averages for fine days both daily sunshine >10 hr and daily wind speed <3 m s$^{-1}$ (11 days in 2018, 10 days in 2019). Right column (b and d): averages for cloudy days all daily sunshine <5 hr, daily wind speed <3 m s$^{-1}$, and daily rain <3 mm (4 days in 2018. 7 days in 2019).

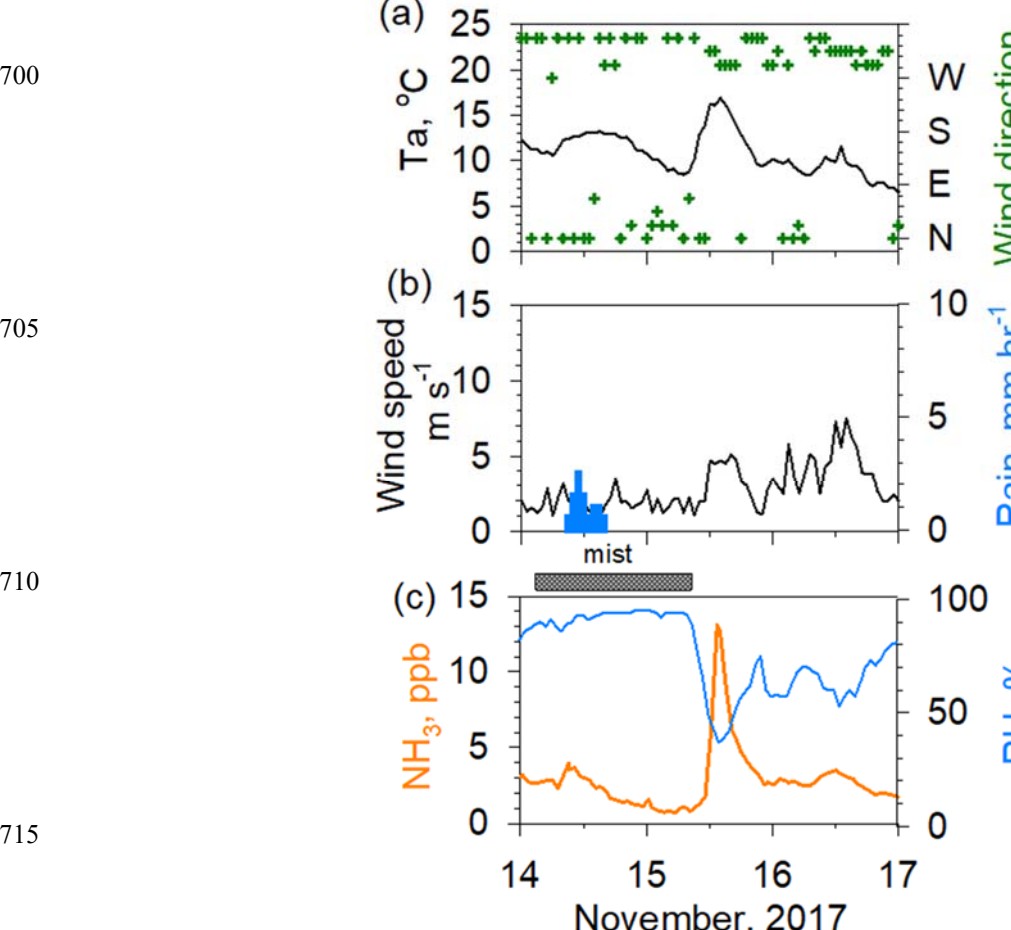

Figure 6 Effects of rain–mist events on ambient $NH_3$ concentrations during 14–17 November 2017: a) temperature (left axis) and wind direction (right axis); b) windspeed and rain rate; c) $NH_3$ and relative humidity.

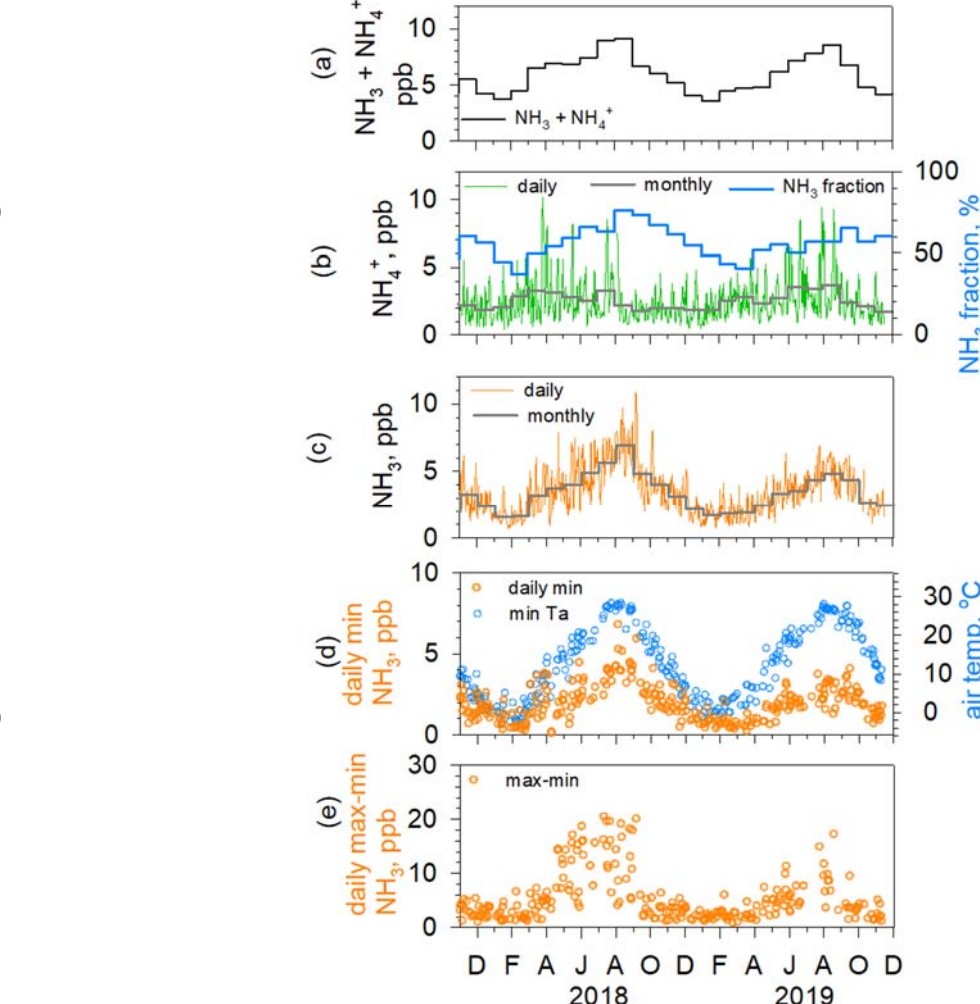

Figure 7 a) Monthly $NH_3 + NH_4^+$ concentrations. b) Daily (thin green) and monthly (thick gray) $NH_4^+$ concentrations with $NH_3$ fraction (thick blue) to $NH_3 + NH_4^+$ concentration. c) Daily (thin orange) and monthly (thick gray) $NH_3$ concentrations. d) Daily minimum $NH_3$ concentrations (orange circle) and minimum air temperature (blue circle) for days of both average relative humidity <70% and daily average wind speed <3 m s$^{-1}$. e) Range of diurnal $NH_3$ concentrations (max-min: orange circle) for days of both average relative humidity <70% and daily average wind speed <3 m s$^{-1}$.

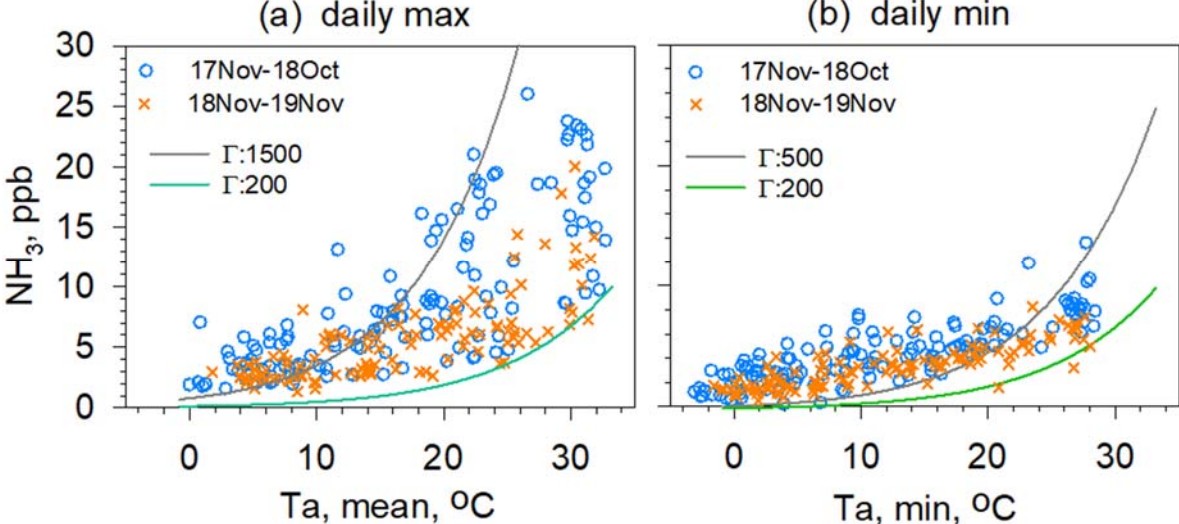

Figure 8 a) Scatter plot showing the maximum NH$_3$ concentration and average air temperatures for days of both
average relative humidity below 70% and daily average wind speed below 3 m s$^{-1}$. b) Scatter plot showing daily
minimum NH$_3$ concentration and minimum air temperature for days of both average relative humidity below 70%
and daily average wind speed below 3 m s$^{-1}$. Green and gray curves show compensation points for temperature
using $\Gamma$ values shown in the panels.

(a) (b)

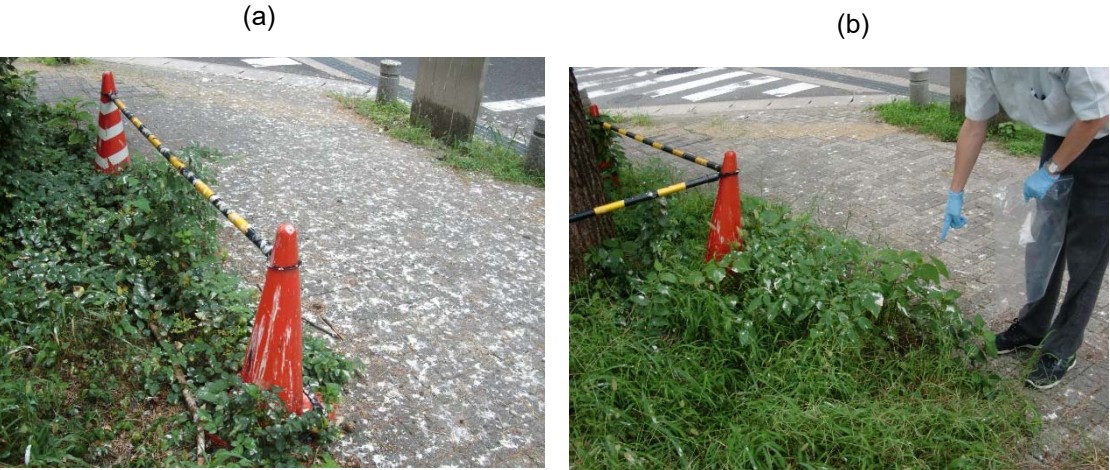

Appendix Photograph 1 Crow droppings (photographs taken at the same place on (a) 28 July, 2018; (b) 17 October, 2019) in front of the Environmental Studies Hall of Nagoya University.

795