# Peer review of "Measurement report: Short-term variation of ammonia concentrations in an urban area increased by mist evaporation and emissions from a forest canopy with bird droppings"

_Atmospheric Chemistry and Physics, 2020_

## Referee Comment (RC1) · Anonymous Referee #1 · 20 May 2020

The author reports two years of NHx and NH4+ data collected in Nagoya, Japan, and uses it to infer local sources of NH3, such as traffic, plant stomata, soil pore water, and bird droppings. Observations of NH3 are consistent with other studies (e.g., daytime maximum, strong seasonal variations), and data are presented in a fairly clear manner. The most novel part of the manuscript is the finding that bird droppings could be a relevant local source of NH3 in urban areas. Although the manuscript is fairly well written, phrasing and grammar could be improved throughout the manuscript. Portions of the data analysis and discussion could be expanded to improve the manuscript, as noted

below. Nonetheless, this measurement report contains valuable insight for understanding urban ammonia sources, and publication is recommend once the comments below are addressed.

General Comments:

Most of the data analysis only considers parts of the data set (e.g., July/December 2018, and RH < 70% when wind speed < 3 m/s). A more holistic look at the entire data set might give additional insight on various sources.

For example, the morning NH3 peak ∼2-4 hours after sunrise is decoupled from the maximum ambient temperature which is inconsistent with bi-directional exchange (i.e., stomata and soil) driving NH3 emissions, since these emissions should peak with temperature. Is it possible the lack of a coincident peak of NH3 and temperature is caused by enhanced vertical mixing (i.e., dilution) later in the day?

Furthermore, examining days with presumed surface wetness (i.e., RH > 70%) might provide insight on whether or not the morning peak in Fig. 4 (top left) is related to evaporation of surface wetness. In other words, the different peak times for NH3 and temperature, as well as the impact of surface wetness evaporation should be explored further.

Specific Comments:

Line 108 – what was the measurement height above the ground?

Lines 108 to 124 – what is the approximate residence time of the air sample, and distance it travels from the inlet, before it comes into contact with the water droplets (i.e., is dissolved)? Is it possible that some relevant fraction of NH3 partitions to the surface of the sampling inlet, which could desorb later at high temperatures and/or lower NH3 concentrations? In other words, has collection efficiency of the system been tested?

Lines 144 to 145 – presumably diurnal variation in wind speed is not as clear in wintertime due to the lack of sea breeze circulations, although the current phrasing implies a direct link between sunlight and wind speed. Recommend rephrasing to clarify that it's not sunlight that's directly impacting wind speed.

Section 3.1 – the analysis focuses on only two months (July and December 2018). Is there a reason that more months weren't considered (e.g., Dec 2017, July 2019, Nov/Jan, June/Aug) when trying to interpret seasonal differences? Considering these additional months would likely make the analysis more representative of the winter/summer seasons.

Section 3.2 – there is a lot of discussion about mist/droplet pH; however, the impact of pH on NH3 release from evaporating mist/droplets is not made clear. It would be helpful to provide a few sentences explicitly stating how NH3 emissions from droplets are impacted by pH.

Lines 226 to 227 – the assumption is that the air being sampled before sunrise under low wind conditions reflects local sources. However, is it possible that the nocturnal boundary layer is sufficiently shallow during these times, such that the sampling inlet on the 7th floor is above the nocturnal boundary and is decoupled from surface sources?

Lines 303 to 306 – the description of crow abundance and behaviour is very anecdotal. A more detailed description on what is meant by terms like "visual impression" and "rarely observed" would be useful.

Line 322 – is this a unit conversion error (3.4 mol m-2 day-1 to 100 mmol m-2 month-1)?

Figure 1b – please add a scale for distance.

---

## Referee Comment (RC2) · Anonymous Referee #2 · 30 Jun 2020

The author presents the analysis of 2 year hourly NHx data in Nagoya, Japan suggesting trends in ambient NH3 are due to mist evaporation and the N input of bird dropping into the surrounding vegetation. This manuscript aims at better characterizing NH3 emission sources in urban areas, which is needed. This study shows the increasing importance of bird guano as a significant source of NH3 is also true for urban areas where high populations of fowl can congregate. The long-term measurements of NHx for this region is valuable and the analysis is sound but somewhat incomplete. I would recommend publishing this manuscript after some revisions.

[Figure]

Major Comments

What was the measurement height of NHx? Could the repeated morning increase also be due to the increase in the boundary layer height?

The correlation to NOx measurements is useful in getting a sense of how much vehicles are contributing to total NH3 emissions. Since, as the author mentions, NH3 can easily react to form NH4, do the correlations of NOx to NHx look similar? since NHx is a better-conserved tracer for all emitted NH3.

Related to the comment above, the discussion around seasonal and interannual variations is focused on NH3, which may underestimate the impact of local sources if any NH3 is reacted to form NH4 - especially at the time resolution of the measurements. Since the study includes measurements of NH3 and NH4 (not an easy task) for such an extensive period, what do the variations in total NHx (and what % NHx is NH3) look like? are the conclusions the same?

In section 3.2, there is a brief mention of some of the other chemical components in rain. The NH4 content is reported later in section 3.3. Based on the reported pH and assuming the rain and mist content have similar NH4 content, could the fraction of NH3 emitted from mist evaporation be calculated using the expression for dew? Does this match the observed increase in NH3?

There is no discussion on the role of cuticular deposition, which is generally represented as a constant NH3 sink (Sutton et al. 1995, 1998; Flechard et al. 1999) in forest canopies. From the photograph of bird dropping, there also appears to be an increase in vegetation. The increase in leaf surface area could potentially increase the amount of NH3 dry deposited to the cuticles, also reducing overall ambient NH3 concentrations. The author discusses the potential difference in N inputs between years and is correct that both soil and leaf stoma can act as reservoirs. Can the author also comment on changes in the local NH3 sinks between years as well that would also affect the overall ambient NH3 concentrations?

Work by Decina S.M. et al (Ponette-González A.G., Rindy J.E. (2020) Urban Tree Canopy Effects on Water Quality via Inputs to the Urban Ground Surface. In: Levia D., Carlyle-Moses D., Iida S., Michalzik B., Nanko K., Tischer A. (eds) Forest-Water Interactions. Ecological Studies (Analysis and Synthesis), vol 240. Springer, Cham) shows vegetation in urban environments tend to concentrate pollutants and input them into the ground surface. The author makes an important point that for NH3 this exchange is bi-directional. The discussion around comparing the estimated compensation point of soil/leaf surface with ambient NH3 concentration does not account for the transfer velocity that ultimately determines the magnitude (and likelihood) of the exchange. Massad et al. (2010) provide a detailed description of this parameter. Would the conclusions be the same when accounting for the transfer velocity?

Minor Suggested Edits

The article would benefit from another round of general grammar and writing edits.

Include dates in Figure captions: Figure 5. Impact of the rain–mist event on the ambient NH3 concentrations from 14 to 17 November 2017.

Measurements highlighting the importance of bird guano as a significant NH3 source is relatively recent, the authors should also include the work of Croft, B.; Wentworth, G. R.; Martin, R. V.; Leaitch, W. R.; Murphy, J. G.; Murphy, B. N.; Kodros, J. K.; Abbatt, J. P. D.; Pierce, J. R. Contribution Of Arctic Seabird-Colony Ammonia To Atmospheric Particles And Cloud-Albedo Radiative Effect. Nature Communications 2016, 7, 13444.

Work by Hrdina, A. H. I.; Moravek, A.; Schwartz-Narbonne, H.; Murphy, J. G. Summertime Soil-Atmosphere Ammonia Exchange In The Colorado Rocky Mountain Front Range Pine Forest. Soil Systems 2019, 3(1) (Special Issue "Formation and Fluxes of Soil Trace Gases") also supports the dynamic range of soil emission potentials chosen by the author.

---

## Author Comment (AC1) · 23 Aug 2020

**Author's response to Referee #1**

**General Comments to the Author**

The author reports two years of NHx and $NH_4^+$ data collected in Nagoya, Japan, and uses it to infer local sources of $NH_3$, such as traffic, plant stomata, soil pore water, and bird droppings. Observations of $NH_3$ are consistent with other studies (e.g., daytime maximum, strong seasonal variations), and data are presented in a fairly clear manner. The most novel part of the manuscript is the finding that bird droppings could be a relevant local source of $NH_3$ in urban areas. Although the manuscript is fairly well written, phrasing and grammar could be improved throughout the manuscript. Portions of the data analysis and discussion could be expanded to improve the manuscript, as noted below. Nonetheless, this measurement report contains valuable insight for understanding urban ammonia sources, and publication is recommended once the comments below are addressed.

General Comments:

Most of the data analysis only considers parts of the data set (e.g., July/December 2018, and RH < 70% when wind speed < 3 m/s). A more holistic look at the entire data set might give additional insight on various sources.

For example, the morning $NH_3$ peak ~2-4 hours after sunrise is decoupled from the maximum ambient temperature which is inconsistent with bi-directional exchange (i.e., stomata and soil) driving $NH_3$ emissions, since these emissions should peak with temperature. Is it possible the lack of a coincident peak of $NH_3$ and temperature is caused by enhanced vertical mixing (i.e., dilution) later in the day?

Furthermore, examining days with presumed surface wetness (i.e., RH > 70%) might provide insight on whether or not the morning peak in Fig. 4 (top left) is related to evaporation of surface wetness. In other words, the different peak times for $NH_3$ and temperature, as well as the impact of surface wetness evaporation should be explored further.

**Response:**

I thank anonymous Referee #1 for constructive comments exploring the cause of the morning $NH_3$ peak described in the manuscript. I asked a native-English proofreader to review the revised manuscript, particularly for clarity in preference to subjective style. Modified words and sentences in the text have been highlighted as yellow in the manuscript.

To cover a longer span and details of seasons, I modified Fig. 4 including the monthly average of diurnal variations for every 3 months from December 2017 through September 2019. I also added a new Fig. 5 to discuss the effects of direct sunshine on the morning peak. Related to the new figures, I have added additional discussion of the boundary layer height (BLH). Unfortunately, data related to BLH were not available for this study. I agree that dilution effects

during daytime reduce the $NH_3$ concentration. Diurnal variation of $NH_3$ concentrations for cloudy days shows a coincident peak with air temperature. In addition, strong effects of the inversion layer on $NH_3$ concentration must be limited for days of calm winds in winter, when the morning peak was absent.

**Specific comments**

**Comment #1:**

Line 108 – what was the measurement height above the ground?

**Response:**

The inlet height was ca. 26 m above the ground. I have added this explanation to the text and to the caption of Figure 1d.

**Comment #2:**

Lines 108 to 124 – what is the approximate residence time of the air sample, and distance it travels from the inlet, before it comes into contact with the water droplets (i.e., is dissolved)? Is it possible that some relevant fraction of $NH_3$ partitions to the surface of the sampling inlet, which could desorb later at high temperatures and/or lower $NH_3$ concentrations? In other words, has collection efficiency of the system been tested?

**Response:**

The distance from the end of the denuder glass tube to the mixing point of water droplets is approximately 10 cm. Because the sample air is transported through the PTFE tube (id: 3 mm) at a flow rate of 1 L $min^{-1}$, the residence time from the end of the denuder to the mixer is ca. 0.04 s, which means that water droplets immediately contact with the sample air after leaving the denuder glass tube. The collection efficiency of the system was higher than 95% for the condition employed in this study. I added these points to the revised text.

**Comment #3:**

Lines 144 to 145 – presumably diurnal variation in wind speed is not as clear in winter time due to the lack of sea breeze circulations, although the current phrasing implies a direct link between sunlight and wind speed. Recommend rephrasing to clarify that it's not sunlight that's directly impacting wind speed.

**Response:**

I revised this part of the text as suggested.

**Comment #4:**

Section 3.1 – the analysis focuses on only two months (July and December 2018). Is there a reason that more months weren't considered (e.g., Dec 2017, July 2019, Nov/Jan, June/Aug) when trying to interpret seasonal differences? Considering these additional months would likely make the analysis more representative of the winter/summer seasons.

**Response:**
To present more data, I renewed Fig. 4 to include the monthly average of diurnal variations for every 3 months from December 2017 through September 2019.

**Comment #5:**
Section 3.2 – there is a lot of discussion about mist/droplet pH; however, the impact of pH on $NH_3$ release from evaporating mist/droplets is not made clear. It would be helpful to provide a few sentences explicitly stating how $NH_3$ emissions from droplets are impacted by pH.

**Response:**
Additional explanations of chemical composition and pH were added to the manuscript, highlighting effects on $NH_3$ emission after the evaporation of mist droplets.

**Comment #6:**
Lines 226 to 227 – the assumption is that the air being sampled before sunrise under low wind conditions reflects local sources. However, is it possible that the nocturnal boundary layer is sufficiently shallow during these times, such that the sampling inlet on the 7th floor is above the nocturnal boundary and is decoupled from surface sources?

**Response:**
Although a detailed time evolution of the nocturnal boundary layer was not known for the site, the sampling inlet (26 m above the ground) is presumed to be located well within the boundary layer. I added discussion specifically related to the controlling factors of $NH_3$ concentration related to the boundary layer height, dry deposition of $NH_3$, and local emissions.

**Comment #7:**
Lines 303 to 306 – the description of crow abundance and behavior is very anecdotal. A more detailed description on what is meant by terms like "visual impression" and "rarely observed" would be useful.

**Response:**
I modified the sentence related to appendix photograph 1 and deleted the sentence that used

"rarely observed" from the revised manuscript.

**Comment #8:**

Line 322 – is this a unit conversion error (3.4 mol m$^{-2}$ day$^{-1}$ to 100 mmol m$^{-2}$ month$^{-1}$)?

**Response:**

I corrected the unit of deposition (3.4 mmol m$^{-2}$ day$^{-1}$).

**Comment #9:**

Figure 1b – please add a scale for distance.

**Response:**

I added a scale for distance in Figure 1b to the revised manuscript.

---

## Author Comment (AC2) · 23 Aug 2020

**Author's response to Referee #2**

**General Comments to the Author**

The author presents the analysis of 2 year hourly NHx data in Nagoya, Japan suggesting trends in ambient $NH_3$ are due to mist evaporation and the N input of bird dropping into the surrounding vegetation. This manuscript aims at better characterizing $NH_3$ emission sources in urban areas, which is needed. This study shows the increasing importance of bird guano as a significant source of $NH_3$ is also true for urban areas where high populations of fowl can congregate. The long-term measurements of NHx for this region are valuable and the analysis is sound but somewhat incomplete. I would recommend publishing this manuscript after some revisions.

**Response:**

I thank anonymous Referee #2 for valuable comments on the overall clarity of the intended message conveyed by the manuscript. We have improved the manuscript according to comments from reviewers. Modified words and sentences have been highlighted as yellow in the revised manuscript.

**Major comment #1:**

What was the measurement height of NHx? Could the repeated morning increase also be due to the increase in the boundary layer height?

**Response:**

The inlet height was 26 m above the ground. I added this point to the manuscript and the caption of Fig. 1d. Referee #1 also pointed out the aspects of time change of the boundary layer height. I agree that the dilution effect during daytime reduces $NH_3$ concentration. Unfortunately, no micrometeorological observation was conducted during this study. Instead, I added some discussion about the controlling factors of $NH_3$ concentration related to boundary layer height, dry deposition of $NH_3$, and local emissions.

**Major comment #2:**

The correlation to NOx measurements is useful in getting a sense of how much vehicles are contributing to total $NH_3$ emissions. Since, as the author mentions, $NH_3$ can easily react to form $NH_4^+$, do the correlations of NOx to NHx look similar? since NHx is a better-conserved tracer for all emitted $NH_3$.

**Response:**

As the lowest panels in Figs. 2 and 3 show, temporal variation of $NH_3$ concentration did not correlate well with $NOx$ concentration. Similarly, $NHx$ and $NOx$ showed no good correlation. I added more discussion on this point using supplemental Figure 1, which shows scatter plots between $NHx$ and $NOx$ as well as $CO$ and $NOx$ in December 2018 and 2019, respectively.

**Major comment #3:**

Related to the comment above, the discussion around seasonal and interannual variations is focused on $NH_3$, which may underestimate the impact of local sources if any $NH_3$ is reacted to form $NH_4^+$ - especially at the time resolution of the measurements. Since the study includes measurements of $NH_3$ and $NH_4^+$ (not an easy task) for such an extensive period, what do the variations in total $NHx$ (and what % $NHx$ is $NH_3$) look like? are the conclusions the same?

**Response:**

I added more data and discussion on $NH_4^+$ and $NHx$, such as average diurnal variation of $NH_4^+$ in new Fig. 4, seasonal variation of $NH_4^+$, and the fraction of $NH_3$ to $NHx$ in the new Fig. 7.

**Major comment #4:**

In section 3.2, there is a brief mention of some of the other chemical components in rain. The $NH_4^+$ content is reported later in section 3.3. Based on the reported pH and assuming the rain and mist content have similar $NH_4^+$ content, could the fraction of $NH_3$ emitted from mist evaporation be calculated using the expression for dew? Does this match the observed increase in $NH_3$?

**Response:**

According to an acid rain report by Nagoya City Institute of Environmental Sciences (NCIES), the volume weighted mean pH of the weekly collected rain samples from 13–20 November, 2017 was 6.00. Based on major ionic data reported for the sample, Frac ($NH_4^+$) proposed in Wentworth et al. (2016) was estimated as 0.14, which suggests the possibility of $NH_3$ evaporation. However, rain was observed twice on the 14th (9 mm) and 18th (16.5 mm) during the sampling period. Unfortunately, the chemical composition of individual rain was not known. In addition, the amount of mist droplets of the event was unavailable. Therefore, the amount of $NH_3$ evaporated from mist droplets could not be estimated. The statement of "A similar rapid $NH_3$ increase up to 15 ppb during 4 hr" was related to another event which occurred in December 11, 2015. For the event in 2015, detailed data about the rain composition were collected; we were able to use it. In the present manuscript, the date of the event in 2015 was added. The description of sea salt and $Ca^{2+}$ for the rain in 2015 was deleted to avoid confusion. In addition, the explanation of NCIES data was rewritten as presented above.

**Major comment #5:**

There is no discussion on the role of cuticular deposition, which is generally represented as a constant $NH_3$ sink (Sutton et al. 1995, 1998; Flechard et al. 1999) in forest canopies. From the photograph of bird dropping, there also appears to be an increase in vegetation. The increase in leaf surface area could potentially increase the amount of $NH_3$ dry deposited to the cuticles, also reducing overall ambient $NH_3$ concentrations. The author discusses the potential difference in N inputs between years and is correct that both soil and leaf stoma can act as reservoirs. Can the author also comment on changes in the local $NH_3$ sinks between years as well that would also affect the overall ambient $NH_3$ concentrations?

**Response:**

I agree with the importance of cuticular deposition on $NH_3$ concentration. Brief discussion of the importance of cuticular deposition and its variation in 2018 and 2019 were added to the revised manuscript.

**Major comment #6:**

Work by Decina S.M. et al (Ponette-González A.G., Rindy J.E. (2020) Urban Tree Canopy Effects on Water Quality via Inputs to the Urban Ground Surface. In: Levia D., Carlyle-Moses D., Iida S., Michalzik B., Nanko K., Tischer A. (eds) Forest-Water Interactions. Ecological Studies (Analysis and Synthesis), vol 240. Springer, Cham) shows vegetation in urban environments tend to concentrate pollutants and input them into the ground surface. The author makes an important point that for $NH_3$ this exchange is bi-directional.

**Response:**

I added relevant discussion with this reference to the last part of section 3.3.

**Major comment #7:**

The discussion around comparing the estimated compensation point of soil/leaf surface with ambient $NH_3$ concentration does not account for the transfer velocity that ultimately determines the magnitude (and likelihood) of the exchange. Massad et al. (2010) provide a detailed description of this parameter. Would the conclusions be the same when accounting for the transfer velocity?

**Response:**

I agree that the ambient $NH_3$ concentration depends on various parameters including the transfer

velocity. Data related to flux estimation were not available for this study. Further study including flux estimations is necessary to evaluate the impact of bird droppings on urban $NH_3$ emissions. Nonetheless, important suggestions can be made for potential sources at the site. I added need of further data to evaluate $NH_3$ exchange.

**Minor suggested edits #1:**
The article would benefit from another round of general grammar and writing edits.

**Response:**
I asked an experienced native-English speaking proofreader for further improvement and clarification of the text of the revised manuscript. Although preferences for style can be subjective, we hope that the changes will clarify all points for all readers.

**Minor suggested edits #2:**
Include dates in Figure captions: Figure 5. Impact of the rain–mist event on the ambient $NH_3$ concentrations from 14 to 17 November 2017.

**Response:**
The caption of the new Fig. 6 (previously Fig. 5) was modified as the reviewer has suggested.

**Minor suggested edits #1:**
Measurements highlighting the importance of bird guano as a significant $NH_3$ source is relatively recent, the authors should also include the work of Croft, B.; Wentworth, G. R.; Martin, R. V.; Leaitch, W. R.; Murphy, J. G.; Murphy, B. N.; Kodros, J. K.; Abbatt, J. P. D.; Pierce, J. R. Contribution Of Arctic Seabird-Colony Ammonia To Atmospheric Particles And Cloud-Albedo Radiative Effect. Nature Communications 2016, 7, 13444.
Work by Hrdina, A. H. I.; Moravek, A.; Schwartz-Narbonne, H.; Murphy, J. G. Summertime Soil-Atmosphere Ammonia Exchange In The Colorado Rocky Mountain Front Range Pine Forest. Soil Systems 2019, 3(1) (Special Issue "Formation and Fluxes of Soil Trace Gases") also supports the dynamic range of soil emission potentials chosen by the author

**Response:**
These references were cited in the revised manuscript.

Supplement Figure

[Figure]

Supplement Figure 1 Scatter plots between NOx and NHx (NH$_3$ + NH$_4^+$), and NOx and CO concentrations.
Upper row, December, 2017; lower row, December, 2018.

---

## Editor Decision (ED1)

**Scientific comments by the editor**

1) l. 145/6: Please clarify what you mean by 'Diurnal variation of wind speed was also unclear ...' . Do you mean 'Diurnal wind speed was variable in winter'?

2) l. 224: Was the pH reported in the NCIES acid rain report indeed given with two digits after the comma? Or was it rather just a pH of 6?

3) l. 267: Instead of saying 'we discuss…. later' please refer to the respective Section

4) l. 374/5 (last sentence of Section 3): Please add a reference for this, unless it is a conclusion from your current study (?).
5) Data availability: See the guidelines on data availability
https://www.atmospheric-chemistry-and-physics.net/about/data_policy.html
All data need to be made available according to the EGU data policy. This is of highest importance in particular in Measurement Reports, because their focus is on the reported data as they include a less thorough analysis and synthesis as required in regular Research Articles. Thus, a statement 'upon request from the author' is not sufficient.
6) Figures: Label all figures panels with a, b, c etc and expand the figure captions that all panels are described separately, e.g.
> Figure 2 a) Temperature (left axis) and relative humidity (right axis); b) Wind speed, rain rate and wind direction; c) $NH_3$ and $NO_2$ mixing ratios

**Technical comments:**

- throughout the manuscript: replace NHx by $NH_x$

- l. 39:  'the major precursor' replace by 'a major precursor'

- l. 87: 'air pollution concentration' should be 'concentration of air pollutants' or 'air pollution levels'

- l. 113: has $NH_4^+$(p) be defined ? ( I assume (p) refers to 'particulate?)

l. 171/2: do you mean 'during rush hour between 7 and 9 am'?

- l. 198: replace 'evidence related to' by 'trend with '

- l. 177 and l. 209: ' a mirror of' sounds awkward; it can be simply replaced by 'due to'

- l. 221: replace 'similar' by 'similarly'

- l. 225:  'with' can be removed

- l. 257: add 'ppb' after 2.8

l. 346: replace 'chickens' by 'chicken' (singular and plural forms are identical)

l. 60: replace 'converted as' by 'converted to'

---

## Author Response (AR2)

Dear Professor Ervens:

Thank you for your comments. Please find enclosed the revised manuscript in 'track change mode' for "Measurement report: Short-term variation of ammonia concentration in an urban area: contributions of mist evaporation and emissions from a forest canopy with bird droppings" by K. Osada (acp-2020-244).

**Scientific comments by the editor**

1) l. 145/6: Please clarify what you mean by 'Diurnal variation of wind speed was also unclear ...' . Do you mean 'Diurnal wind speed was variable in winter'?

**Response:**

I modified the sentence as suggested.

2) l. 224: Was the pH reported in the NCIES acid rain report indeed given with two digits after the comma? Or was it rather just a pH of 6?

**Response:**

It was 6.0. I corrected this in the manuscript.

3) l. 267: Instead of saying 'we discuss…. later' please refer to the respective Section

**Response:**

I modified as the following: We discuss more details related to this point later using Fig. 8b.

4) l. 374/5 (last sentence of Section 3): Please add a reference for this, unless it is a conclusion from your current study (?).

**Response:**

I agree with the editor and have deleted the last sentence.

5) Data availability: See the guidelines on data availability https://www.atmospheric-chemistry-and-physics.net/about/data_policy.html

All data need to be made available according to the EGU data policy. This is of highest importance in particular in Measurement Reports, because their focus is on the reported data as they include a less thorough analysis and synthesis as required in regular Research Articles. Thus, a statement 'upon request from the author' is not sufficient.

**Response:**

I added a supplemental data table of NHx data and modified as the following: The NHx data used for analysis are available in the Supplement, and the data are also available upon request from the corresponding author

6) Figures: Label all figures panels with a, b, c etc and expand the figure captions that all panels are described separately, e.g.

Figure 2 a) Temperature (left axis) and relative humidity (right axis); b) Wind speed, rain rate and wind direction; c) $NH_3$ and $NO_2$ mixing ratios

**Response:**

I modified figures as suggested.

Technical comments:

- throughout the manuscript: replace NHx by NHx
- l. 29: 'the major precursor' replace by 'a major precursor'
- l. 87: 'air pollution concentration' should be 'concentration of air pollutants' or 'air pollution levels'
- l. 113: has $NH_4^+(p)$ be defined ? ( I assume (p) refers to 'particulate?)

l. 171/2: do you mean 'during rush hour between 7 and 9 am'?

- l. 198: replace 'evidence related to' by 'trend with '
- l. 177 and l. 209: ' a mirror of' sounds awkward; it can be simply replaced by 'due to'
- l. 221: replace 'similar' by 'similarly'
- l. 255: 'with' can be removed
- l. 257: add 'ppb' after 2.8

l. 346: replace 'chickens' by 'chicken' (singular and plural forms are identical)

l. 360: replace 'converted as' by 'converted to'

**Response:**

I corrected these as suggested.

[revised manuscript text omitted]

---

## Author Response (AR3)

Dear Professor Ervens:

Thank you for your comments. Please find enclosed the revised manuscript in 'track change mode' for "Measurement report: Short-term variation of ammonia concentrations in an urban area increased by mist evaporation and emissions from a forest canopy with bird droppings" by K. Osada (acp-2020-244).

**Comments by the editor**

As mentioned in my previous comments, it is mandatory in particular for Measurement Reports that ALL data and materials underlying reported findings are reported. I could only find the NHx data in your supplement. I strongly suggest that you upload the other data that are displayed in Figures 2-8 to a suitable repository (see example links below) or make them easily accessible otherwise.

**Response:**

NHx and other data used in Figs 2–8 are available at NAGOYA Repository: http://hdl.handle.net/2237/00032615. I added this information and deleted daily average of NHx data in the supplement file.

I hope that the revised manuscript is now suitable for publication. I look forward to hearing from you at your earliest convenience.

Yours sincerely,

Kazuo Osada

[revised manuscript text omitted]